# Estimated Glomerular Filtration Rate in Chronic Kidney Disease: A Critical Review of Estimate-Based Predictions of Individual Outcomes in Kidney Disease

**DOI:** 10.3390/toxins14020127

**Published:** 2022-02-08

**Authors:** Lajos Zsom, Marianna Zsom, Sohail Abdul Salim, Tibor Fülöp

**Affiliations:** 1Fresenius Medical Care, Cegléd Dialysis Center, Törteli u 1-3, 2700 Cegléd, Hungary; 2Department of Medicine, St. Rókus Hospital, Rókus u 10, 6500 Baja, Hungary; zsom.mariann@gmail.com; 3Department of Medicine, Division of Nephrology, University of Mississippi, 2500 N State St., Jackson, MS 39216, USA; sohail3553@gmail.com; 4Department of Medicine, Division of Nephrology, Medical University of South Carolina, 96 Jonathan Lucas Street, MSC 629, CSB 822, Charleston, SC 29425, USA; 5Medicine Services, Ralph H. Johnson VA Medical Center, 109 Bee St., Charleston, SC 29401, USA

**Keywords:** chronic kidney disease, end-stage kidney disease, glomerular filtration rate, eGFR

## Abstract

Chronic kidney disease (CKD) is generally regarded as a final common pathway of several renal diseases, often leading to end-stage kidney disease (ESKD) and a need for renal replacement therapy. Estimated GFR (eGFR) has been used to predict this outcome recognizing its robust association with renal disease progression and the eventual need for dialysis in large, mainly cross-sectional epidemiological studies. However, GFR is implicitly limited as follows: (1) GFR reflects only one of the many physiological functions of the kidney; (2) it is dependent on several non-renal factors; (3) it has intrinsic variability that is a function of dietary intake, fluid and cardiovascular status, and blood pressure especially with impaired autoregulation or medication use; (4) it has been shown to change with age with a unique non-linear pattern; and (5) eGFR may not correlate with GFR in certain conditions and disease states. Yet, many clinicians, especially our non-nephrologist colleagues, tend to regard eGFR obtained from a simple laboratory test as both a valid reflection of renal function and a reliable diagnostic tool in establishing the diagnosis of CKD. What is the validity of these beliefs? This review will critically reassess the limitations of such single-focused attention, with a particular focus on inter-individual variability. What does science actually tell us about the usefulness of eGFR in diagnosing CKD?

## 1. Introduction: History of a Concept with Assumptions Conveniently Forgotten

### 1.1. The Concept of Glomerular Filtration Rate and Its Estimation by Creatinine Clearance

Glomerular filtration rate (GFR) was originally introduced to estimate glomerular function by calculating the amount of fluid filtered through the renal glomeruli per unit of time. It is not a measure of single-nephron glomerular function; rather, it is a measure reflecting the summation of the filtration of all glomerular capillaries in the human kidney [1]. When a solute is freely filtered through the glomeruli and is neither reabsorbed nor secreted by tubules, as in the case of inulin, then the clearance of that solute may be used to measure GFR. Thus, GFR has been determined by finding the volume of blood glomeruli clear of insulin per minute and is calculated by the formula urine concentration times urine flow per plasma concentration [2,3]. In clinical practice, creatinine is substituted for inulin as creatinine is present naturally in the body so it does not need to be injected like inulin [4,5]. However, creatinine is not an ideal marker for estimating GFR due to its tubular secretion that increases through the course of renal disease; the more advanced the disease the larger the ratio of secreted to freely filtered creatinine [6]. In addition, creatinine being a waste product of protein metabolism in muscles, one must assume that creatinine clearance as a true reflection of intrinsic renal pathology may theoretically depend on the condition of a steady state with a constant and stable generation of creatinine in the muscle unaffected by catabolism (1), no changes in muscle activity (2), or dietary influences (3). It may also be assumed that creatinine should have a stable distribution with a relatively constant concentration in the serum (4) and adequate delivery to the glomerular capillaries using the so-called one compartment model (5) necessitating stable cardiovascular status and good vascular supply to the kidneys with the absence of acute changes in cardiac output or hydration status, current administration of non-steroid anti-inflammatory drugs (NSAID) or of any other medication acutely affecting renal blood flow (RBF) including blood pressure lowering agents, especially angiotensin converting enzyme (ACE) inhibitors or angiotensin II type 1 receptor blockers (ARBs). Functionally, the human kidneys can be simplified into two conceptual compartments—one of a filter and the other one the repressor; however, it is only the latter one that is energy expensive in terms of O_2_ utilization. Hence, reducing GFR without reducing overall RBF would confer a better overall O_2_ supply to the tubuli and the medulla.

It is important to recognize that creatinine clearance as a reflection of decreased filtration due to actual renal disease is dependent on all these assumptions and that especially if the autoregulation of the afferent arterioles is affected by chronic disease such as diabetes mellitus, hypertension, congestive heart failure, or many others [7,8], creatinine clearance will be highly variable creating a snapshot effect should creatinine clearance be measured, like in many studies, only a few times a year.

### 1.2. The Concept of the Estimation of Glomerular Filtration Rate by Estimating the Clearance of a Marker: The Estimation of an Estimation 

Since the calculation of creatinine clearance by a 24-hour urine collection has been cumbersome and is subject to much imprecision during collection, mGFR (measured GFR) as measured by urine collection of filtered creatinine has been largely replaced by eGFR, derived from approximations obtained through large epidemiological studies establishing correlations between serum creatinine and mGFR as influenced by parameters known to modify this relationship including age, sex, race, and others [9,10,11]. One such formula is CKD-EPI [12,13], which was obtained from large cross-sectional studies of patients with or without renal disease where correlation between serum creatinine values and mGFR as measured by clearance of exogenous filtration markers such as iothalamate has been established using the formula GFR = 141 × min (S_cr_/κ, 1)^α^ × max(S_cr_/κ, 1)^−1.209^ × 0.993^Age^ × 1.018 (if female) × 1.159 (if black), accounting for variables of age, sex, and race (black vs. non-black). Yet, in the study populations used for establishing this formula, the elderly, especially subjects above 65 years of age, black subjects, and patients with an actual diagnosis of CKD were largely under-represented [12]. Furthermore, the equation obtained is based on a population-based average ignoring such unique individual factors as muscle mass, body composition, or the presence or absence of steady state; the latter commonly occurring in states of catabolism. The most recent development is the acceptance of a race-free formula in the United States [14]; however, none of these concepts fully consider the slow evolvement of the human body composition and significant changes of diet, lifestyle, and physical activity with the post-industrial era, all potentially impacting the eGFR formula and rendering any formulas less accurate 1 or 2 decades later.

It is commonly accepted that the use of mGFR would provide higher accuracy than most eGFR equations. The performance goal is for eGFR to be within 30% of mGFR values 90% of the time per KDIGO 2012 clinical practice guidelines [15]. Whether any of eGFR equations that use creatinine or Cystatin C accurately reflect kidney function has been debated. Porrini et al. [16] analyzed 70 studies comparing eGFR with mGFR involving 40,000 kidney transplant patients and showed that eGFR often differed from mGFR by ±30% or more, which incorrectly staged CKD in 60% of patients, with eGFR and mGFR showing different rates of GFR decline. Some authors believe that this discrepancy might be partially mitigated by the incorporation of more filtration markers or using their combination to increase predicting value of eGFR. A combination of cystatin C-based eGFR, the inverse of β2-microglobulin concentration and creatinine-based eGFR was found to be a stronger predictor of ESKD than creatinine-based eGFR alone [17]. In addition, cystatin C-based formulae may obviate the need for race-based correction and may more accurately reflect actual GFR in the context of sarcopenia [18]. This may be especially important in the elderly where functional renal decline may be associated with a complex change in metabolic profile [19] including weight loss and changes in serum albumin, C-reactive protein, and a host of other important parameters.

It is obvious that more research is needed to determine the accuracy of GFR estimations; these research efforts may include more precise assessments of novel filtration markers [20,21]. In the meantime, clinicians should keep in mind the limitations of using currently available equations.

An additional real-life problem is the precision of estimating GFR in the elderly leading to a potential bias to diagnose renal disease in the elderly [22,23,24,25] based on eGFR: a relevant clinical problem that may lead to unnecessary procedures and psychological burden especially in the elderly. This issue will be discussed in full detail below.

### 1.3. Summary of Historical Introduction

It is important to realize the potential assumptions implicit in using eGFR to estimate renal function as should be clear from the historical introduction above and further summarized in Table 1. Such assumptions include all of the following: the totality of renal function can be estimated by measuring glomerular function alone (1), renal function can be estimated by measuring the elimination of an ideal substance under steady state circumstances (2), creatinine can be substituted for such an ideal substance (3), ignoring not just its tubular secretion but assuming steady production without variations due to catabolism, or muscle injury (4), lack of major oscillations due to dietary variability (5), one compartment distribution with maintained circulation (6), and perhaps most importantly, disregarding variability in renal blood supply (7).

This is especially misleading when renal autoregulation is impaired and a snapshot view of “GFR” changes may partially reflect the actual hemodynamic situation with hydration status, cardiac output, medications such as diuretics, NSAIDS, ACE inhibitors, ARBs, actual blood pressure, or glomerular pressure all potentially playing a role. Finally, estimation of mGFR by a formula derived from correlations in a population where patients with renal disease, elderly patients, and non-whites were largely under-represented may also lead to a potential error when using eGFR to diagnose CKD in an individual patient.

## 2. Renal Function vs. GFR: Fluid Status, Renal Blood Supply, and GFR Variability

Compared to historical concepts delineated above, in the last several decades it has become abundantly clear that the human kidney has important physiological functions above and beyond glomerular filtration. Do these functions correlate with GFR?

One of the most clinically relevant of the many functions of the human kidney is the regulation of fluid homeostasis [26]. It has been recognized that hypervolemia does correlate with mortality at any level of GFR in part by triggering myocardial remodeling and systemic inflammation [27,28,29,30]. One of the most important aspects of ESKD care is to maintain residual renal function precisely to aid the achievement of euvolemia and to slow down the development of left ventricular hypertrophy, cardiac remodeling, and myocardial stunning in dialysis patients [31,32,33,34]. Hypervolemia is closely related to systemic inflammation [35,36] partly by increasing intestinal permeability resulting in endotoxin absorption [37,38] and the resultant activation of cytokines as observed in both heart failure [39,40,41] and hemodialysis populations [42,43]. Inflammation in turn is closely related to the malnutrition-inflammation syndrome in end-stage renal disease and is significantly correlated with mortality [35,44,45], while it has an inverse correlation with residual renal function in hemodialysis patients [46]. Surely, the chance for hypervolemia is increased with GFR decline, potentially explaining some of the correlation between low GFR and cardiovascular morbidity and mortality [47]. However, changes in fluid status may influence both measured and estimated GFR by multiple mechanisms that may play a role in GFR variability, itself a significant predictor of renal progression and mortality [7,8]. Specifically, it has been shown that a stricter blood pressure control with a decline in GFR may not predict increased risk of ESKD [48] and may confer benefits lowering ESKD risk in certain populations [49] and aggressive diuresis in decompensated heart failure even with initial GFR decline may be associated with better renal outcomes, including the eventual stabilization of renal function [50,51]. It has also been observed that while at any GFR, the degree of proteinuria is a predictor of both renal [52] and cardiovascular [53] outcomes, an improved control of proteinuria, in part by correcting fluid status and using RAS blockade, improves renal outcomes despite initially lowering GFR reducing or in some instances reverting GFR slopes [54]. It appears that in these circumstances snapshot eGFR determinations do not say it all: time dynamics need to be carefully studied as initial GFR decline may actually reflect a beneficial improvement in GFR slopes in the long run in many such patients.

Moreover, in the case of hypervolemia, such as in congestive heart failure, GFR may be less stable as it introduces a highly dietary dependent and partly reversible influence on renal blood supply especially when renal autoregulation is impaired as is often the case with long standing vascular disease [7,8,50,51]. As cardiac output varies due to changing fluid status with varying position on the Starling curve, renal blood supply also changes with alterations in the degree of venous congestion actively influencing GFR [55]. Snapshots of eGFR will then tend to be less precise if they are intended to be used as true reflections of underlying renal pathology or a predictor of renal outcomes.

In fact, it has been shown that GFR variability is an independent predictor for both reaching end-stage kidney disease and death, perhaps at least in part due to its association with cardiovascular disease and hypervolemia as detailed above [7,8].

## 3. Renal Function vs. GFR: Proteinuria

Proteinuria has been part of the definition of CKD [56,57] realizing that at any level of eGFR, the amount of proteinuria is a predictor of end stage kidney disease [58]. In fact, a high degree of proteinuria as assessed by albumin-to-creatinine ratio (>0.3 g/g) at an eGFR of 45–60 seems to predict a much stronger risk for all-cause mortality, progressive decline in renal function, and end-stage kidney failure than an eGFR of 30–45 but with low albuminuria [58,59,60]; in addition, unlike low GFR, persistent proteinuria is nearly always a reflection of intrinsic renal pathology with the degree of proteinuria correlating with clinical outcomes [61]. From the classical studies demonstrating improved renal and cardiovascular outcomes on reducing proteinuria with ACE inhibitors in diabetic nephropathy [62] to more recent studies of similar improvements noted in both non-diabetic kidney disease [63,64] and diabetic nephropathy [65,66,67] treated with SGLT-2 inhibitors, the significance of reducing proteinuria remains widely appreciated by nephrologists, though not always by our non-nephrologist colleagues. Furthermore, a decreased but stable GFR with a low amount of proteinuria may often reflect a primarily non-renal pathology such as is the case from cardiorenal syndrome especially with right sided heart failure to dehydration, renovascular disease, obstructive sleep apnea, obstructive uropathy, and many others. A decreased GFR without a large amount of proteinuria may also predict favorable clinical outcomes during diuresis in congestive heart failure [50,51] or a better control of blood pressure in primary hypertension [48,49]. These observations highlight the main difference between the two parameters present in the definition of CKD: glomerular filtration as assessed by eGFR is highly dependent on extrarenal factors and may be quite variable day-to-day depending on dietary changes, actual cardiac output, the presence and degree of catabolism, exercise and blood pressure control, and is subject to even more variability with impaired autoregulation [7,8]. On the other hand, while persistent proteinuria is clearly dependent on blood pressure control, volume status and glomerular pressure, it nevertheless almost always reflects true intrinsic renal pathology that can be improved by reducing proteinuria [8,61]. This presents a very important challenge in everyday clinical practice: whereas many of our primarily non-nephrologist colleagues tend to equate even a single snapshot of low eGFR with chronic intrinsic renal pathology and make a presumptive diagnosis of CKD when intrinsic renal pathology may not be present, the importance of proteinuria is sometimes neglected [68] leading to inadequate treatment or late diagnosis of progressive renal disease in those who would benefit from early intervention. There is an intrinsic danger in oversimplification as reflected in the famous quote from a prominent physicist: “Every explanation should be made as simple as possible, but no simpler.” Equating an easy-to-obtain laboratory parameter with the presumptive presence of a chronic disease was recognized to be a risk for over-diagnosis [69,70] or late diagnosis [70] of progressive renal disease especially when other risk factors, such as albuminuria, are not considered in the diagnosis and risk assessment.

## 4. GFR Variability and Slopes

Once it is realized that kidney function as estimated by eGFR may be subject to substantial intra-individual variability and susceptible to potentially reversible extra-renal factors, a logical step would be to introduce intra-individual “auto-control” by following up eGFR over time to assess patterns of change. However, depending on intra-individual variability, eGFR snapshots may need to be taken frequently enough in order to establish a clear pattern. In addition, if eGFR slopes, i.e., potentially crude linear approximations of changes in eGFR described using a graphic representation with time as the independent variable, are assessed for a sufficiently long period of time, then age-related changes of eGFR need to be factored in suspecting that with true renal pathology, eGFR slopes would be different compared to patterns reflecting age-related changes alone. For this, one needs a clear idea of how eGFR changes trend in a more elderly population without renal disease. Fortunately, we do have several studies asking precisely these questions, though as we shall see, these studies tend to be either cross-sectional or longitudinal with either too short follow-up and/or too few eGFR values assessed with relatively low frequency. Yet, after a full review of these clinical studies, a pattern seems to emerge.

### 4.1. GFR Variability

Simple logic dictates that the more measurements are taken to assess eGFR in an individual over time, the more intra-individual variability with a non-linear pattern may be uncovered especially if true renal pathology is present. Obviously, two values will exactly determine a linear slope using one of the most basic rules of Euclidian geometry creating a self-fulfilling assumption; on the other hand, a pattern may be more difficult to fit into a linear slope if several measurements are taken revealing true intra-individual variability including potentially reversible acute injury or changes in GFR following temporary alterations in hemodynamic or nutritional status. For eGFR slopes to be meaningful, long follow up and multiple measurements are necessary. This is precisely what some of the research on the issue of GFR variability tells us [71,72]. First, it appears that patients with faster than average renal progression tend to have a less linear course of eGFR decline with more intra-individual variability of eGFR [7,8]. Second, patients with more intra-individual variability seem to be burdened more by diabetes and cardiovascular disease raising the possibility that eGFR variability, at least in this population of patients, may be partly due to variable cardiac output, fluid status, and blood pressure with relatively rapid changes in glomerular pressure secondary to loss of autoregulation [8]. Third, GFR variability is an independent risk factor for poor renal outcomes even after adjusting for CKD stage and eGFR slopes, an unsurprising finding since GFR variability is more common in faster progressors and those with more cardiovascular morbidity [7,8]. This information may be completely lost to us unless eGFR measurements are taken often enough to uncover their true pattern over time avoiding a snapshot effect and creating a comprehensive narrative, adjusting for reversible, unrepresentative drops in eGFR partly due to cardiovascular factors [72].

Yet, this is precisely what happens in many of the studies assessing GFR slopes and outcomes where a linear decline is assumed and conveniently found using annual eGFR measurements over a period of 2–3 years. Nonetheless, when assessed in a sufficiently large number of patients, eGFR slopes do seem to predict renal [72,73], cardiovascular [74], and mortality [75,76] outcomes.

### 4.2. eGFR Slopes with Renal Disease

The predicting value of eGFR slopes for both renal and cardiovascular outcomes, where eGFR slopes were used as a measure of the rate of decline in renal function, have been described in several recent studies. Largely, these studies fall into one of four categories: some focused on the general “healthy” population potentially including incidental lower eGFR with or without established renal pathology [77] (1); others assessed the predicting value of eGFR slopes in patients at risk for kidney disease such as diabetics [72] (2); some studies have been performed in patients with already well-established renal disease [73,78,79] (3); and finally others focused on a healthy elderly population to establish patterns of eGFR decline with ageing but without kidney disease [80] (4). In some of the studies with mixed populations, a faster eGFR decline seems to predict worse renal outcomes [77], a non-independent effect largely due to factors of increased age, female sex, and the presence of comorbidities. In diabetic subjects on ACE inhibitors and diuretics, a steeper slope suggesting a more rapid decline seemed to predict worse renal and cardiovascular outcomes [72]. Patients with steeper slopes also tended to have more proteinuria. This robust finding is significant in the face of reviews pointing out the large margin of error of eGFR vs. mGFR when the former was used for the longitudinal follow-up of renal function in type 2 diabetics [81].

In patients with established CKD, a meta-analysis of 22 cohorts [73] showed evidence that patients with steeper eGFR slopes and lower “baseline” eGFR had a higher risk for subsequent renal replacement therapy: at baseline eGFR levels of 20, 30, and 40 mL/min per 1.73 m^2^, the independent effects of a slope greater than −6 mL/min per 1.73 m^2^ per year predicting worse renal outcomes were robust. Steeper slopes were again associated with higher levels of albuminuria. This analysis also revealed that low eGFR, even with a slope of 0, which suggests stable renal function, substantially increases the risk for end stage kidney disease. Clearly, this may be due to an acute adverse event, renal or extra-renal, having the ability to “tip over” an already low renal function with little renal reserve remaining to compensate leading to the need for initiating renal replacement therapy.

Three recent studies in moderate to severe CKD patients confirmed these findings extending the risk associated with steeper eGFR slopes from renal to non-renal outcomes including all-cause mortality [75,76] and cardiovascular morbidity [74]. Specifically, patients with steeper eGFR slopes had a substantial increase in the risk for both renal and non-renal outcomes. How does this compare with healthy patients with a decline in eGFR solely due to ageing?

### 4.3. GFR Slope with Ageing

With ageing, there is a steady increase in the number of cysts in healthy kidney donors, which itself contributes to decreased GFR [82]. Though focal and global glomerulosclerosis of the obsolete type, interstitial fibrosis with tubular atrophy, and arteriosclerosis increase with ageing kidneys, focal segmental glomerulosclerosis is not a characteristic of ageing; rather, it denotes pathology from other diseases. Rule et al. report that in a study using kidney biopsies from 1203 living donors, the prevalence of nephrosclerosis increased markedly with age, from less than 5% for ages 18–29 years to 30% for 40–49 years and 60% for persons aged 60–69 years [83].

Nephron number and GFR decrease progressively with normal ageing. Denic et al. biopsied 1638 healthy kidney donors, representing a sizable population of healthy humans in the general population with no comorbidities. The results suggested that the mean nephron number as well as the number of non-sclerotic glomeruli decreased with age, while global glomerulosclerosis increased [84]. Thus, even though these findings were not correlated with eGFR decline during these studies, the question arises whether age-related microscopic and macroscopic changes of the kidneys including nephrosclerosis could by themselves decrease GFR and lead to worsened clinical outcomes in certain individuals.

However, there may be a substantial difference in the course and individual risk profile of GFR decline due to ageing alone vs. ageing with specific renal pathology with a potentially significant inter-individual variability.

Specifically, once it is established that the eGFR slope is an independent risk factor for mortality and predicts adverse renal and non-renal outcomes in patients with established chronic renal disease, we need to understand if eGFR decline due to ageing alone confers a similar risk. From a large longitudinal study conducted in Japan [80] on 72521 healthy patients, it seems that the higher the baseline, the steeper the eGFR decline, but at some point, the eGFR decline slows down in the vast majority of patients.

A longitudinal study also from Japan following healthy subjects over 40 years of age for 10 years [77] found that the eGFR slope depended on blood pressure, proteinuria, and baseline eGFR, the latter effect being dependent on the age group. A faster eGFR decline was seen below an initial GFR < 50 mL/min/1.73 m^2^ among subjects younger than 70 years of age; and GFR < 40 mL/min/1.73 m^2^ in the group of ages 70–79. This suggests that in different age groups, different levels of eGFR predict a subsequent decline in renal function as assessed by eGFR, at least in this Japanese population. If these findings could be confirmed, then it appears that (1) the pattern of the age-related decline in GFR in healthy subjects may be different depending on “baseline” eGFR with steeper declines with higher initial eGFRs that may level off in time, and (2) among those below the age-dependent eGFR threshold, a further decline may occur, depending on the control of blood pressure and proteinuria. In either case, it is logical to assume that incidental measurements of eGFR without an assessment of the eGFR slope, proteinuria, blood pressure, age, and co-morbidities should not establish the diagnosis of organic disease in the elderly population. Furthermore, the definition of CKD by snapshots of eGFR may be misleading when population-based results are used for individual assessments of elderly patients; especially in those cases where eGFR slopes may be influenced by the initiation of blood pressure medications including ACE inhibitors or ARBs. Therefore, it appears that rather than focusing on “age-dependent eGFR thresholds”, the diagnosis of CKD in the elderly should be highly individualized taking into account the GFR slope, proteinuria, blood pressure, fluid status, and the use of blood pressure medications potentially altering the context in which the significance of eGFR snapshots is interpreted. Figure 1 delineates a number of potential pitfalls associated with either linear extrapolation of eGFR snapshots or using the absolute value of eGFR as the main predictor of future clinical events in the context of inter-individual variability of CKD progression. 

## 5. Nutrition, Frailty, Metabolic Acidosis, Serum Phosphorus, and Chronic Inflammation

There is no doubt that an association exists between malnutrition and advanced CKD as assessed by eGFR [85]; nevertheless, a considerable inter-individual variety may be present at any given eGFR including even in CKD-5D patients depending on age, comorbidity, presence of metabolic acidosis, anemia, chronic inflammation, hypervolemia, intestinal dysbiosis, and other factors [86]. Metabolic acidosis that may or may not correlate with eGFR in a particular individual has been independently correlated with malnutrition [87], catabolism [88], as well as cardiovascular [89] and renal outcomes including a faster progression of CKD [90]. Even the prevalence of anemia [91] and patterns of metabolic bone [92] disease have been found to be variable at any given eGFR and there is little doubt that both anemia [93] and hyperphosphatemia [94] are important independent predictors of cardiovascular morbidity and mortality.

The argument has been made to consider clinical and nutritional assessments as an important factor determining the timing of renal replacement therapy initiation [95]. It is also clear that a subjective assessment of nutrition with or without an assessment of frailty is one of the most important predictors of clinical outcomes on both hemodialysis and peritoneal dialysis and are superior to assessments of dialysis clearance in this regard [86]. While we may regard eGFR or eKT/v urea as our predominant tool for the assessment of the risk for clinical outcome in CKD especially at the population level, we must be cognizant of the fact that even though these parameters are easy to obtain, easy to document, objective, quantifiable, reproducible, and might seem easy to interpret, they nevertheless do not represent a comprehensive tool in assessing risk and may not correlate well with other independent predictors of outcome, including the subjective global assessment of nutrition [96] in a given individual. On the contrary, a complex “subjective assessment” of clinical status based on strong clinical predictors including nutritional status, inflammation, volume status, serum phosphorus, anemia, the degree of metabolic acidosis, frailty, proteinuria, and perhaps intestinal dysbiosis seems to be the superior strategy in diagnosing intrinsic renal pathology or assessing cardiovascular or mortality risk or even considering the optimal time for initiation of renal replacement therapy.

The main underlying factor is the fact that the human kidney is not merely “a filter”, and its function cannot be assessed by the estimation of “filtering capacity”; rather, the kidney is a multi-function organ with an important role in maintaining fluid homeostasis, serum pH, normal blood cell count, serum phosphorus and vitamin D metabolism, anabolic state without protein losses, a functioning immune system, myocardial function, sympathetic nervous system activity, systemic inflammation, and the protection of the intestinal microbiome, among many others. As mentioned earlier, conceptually, the human kidneys can be split into two conceptual compartments—one of a filter and the other one the repressor of the filtered primary urine; however, it is only the latter one that is energy expensive in terms of O_2_ utilization. Hence, reducing GFR without reducing overall renal blood flow would confer a better overall O_2_ supply of the tubuli and the medulla. Unloading the kidney to prolong renal survival invokes reducing the repressors’ work either via decreasing primary filtrate volume without impairing RBF (like an ACE inhibitor) or reducing the need to recall certain substances (e.g., glucose with SGLT-2 inhibitor) or reducing the burden of new alkaline generation via exogenous NaHCO_3_ supplementation. While the deterioration of some of these functions may correlate on a population level with the decline in eGFR, all these factors need to be individually assessed in a particular patient to get a clear picture of the degree of deterioration of the renal function and its actual consequences.

## 6. Residual Renal Function and Dialysis

Perhaps the most typical example for the consequences of the fallacy of equating renal function with clearance as well as equating actual clinical risk with clearance-based assessments is in the setting of dialysis [31,86,97], which provides a unique picture of what is happening when replacing filtering function alone. First, not all clearances are created equal. It has been convincingly shown that residual renal function (Kr) correlates with survival [98,99,100,101], volume control [102], serum albumin [103], markers of inflammation [46,104], intra-dialytic hypotension [105], nutritional status [103], levels of phosphorus [106] and uremic toxins such as p-cresol, indoxyl sulfate as well as highly protein-bound organic anions [107], whereas dialysis clearance (Kd) does not correlate with the aforementioned parameters or does so under certain circumstances such as correlating with survival on hemodialysis in anuric patients [101]. This means that dialysis and residual clearances cannot be regarded “equal” (Kr ≠ Kd). Second, dialysis clearance as assessed by Kt/V urea ignores the independent effect of dialysis duration (t) on survival [31,108,109,110,111], blood pressure and volume control [108,112,113,114,115,116,117], left ventricular hypertrophy [116,118], serum phosphorus [117], and serum albumin leading to the conclusion that dialysis clearance and dialysis duration cannot be regarded as equivalent factors (K ≠ t). In addition, there is some evidence that dialysis modality may also affect survival and volume control such as in the case of hemodiafiltration vs. high flux dialysis [119,120], though this notion remains admittedly somewhat controversial [121,122].

It appears that dialysis clearance does not say it all: the way we arrive at clearing uremic toxins may affect a number of important clinical parameters, such as volume control and the closely related triad of chronic inflammation, intestinal dysbiosis, and myocardial stunning [123,124,125]. In contrast, nutritional assessments such as the subjective global assessment of nutrition or the malnutrition-inflammation score seem to capture the effect of several such clinical predictors, clearance-based parameters do not [86].

## 7. Conclusions

The human kidney is a multi-function organ with functions ranging from the clearance of metabolic products and medications to the regulation of volume status, hormone secretion, vitamin D metabolism, maintenance of ion and acid-base homeostasis, supporting maintenance of anabolism, immune response, the intestinal microbiome, and cardiovascular health. A decline in filtrating capacity assessed by surrogate markers such as eGFR may predict the risk for end-stage kidney disease, mortality, and cardiovascular morbidity in community based, mainly cross-sectional studies but may not correlate well with the disease process in a particular individual, especially when looked at as an infrequently taken snapshot without fully assessing the underlying context. Yet, this is precisely what is often done.

Ageing, varying blood pressure and volume status, or cardiac output especially with impaired autoregulation and/or ACEI, ARB, or calcineurin inhibitor use, dietary inconsistencies, changes in muscle mass, muscle injury or a catabolic state, the presence or absence of proteinuria, and in certain cases histological or ultrasonographic assessment and eGFR slope should all be individually considered when trying to use eGFR for the diagnosis of CKD. Overall, in order to avoid the pitfall of equating CKD with “Low eGFR Disease”, we think that clinicians should regard eGFR as an initial screening tool and should not arrive at a diagnosis of CKD without carefully assessing the full clinical context. Furthermore, they should carefully adjust their risk assessment in an individual case considering a host of factors including age, frailty, nutritional assessment, metabolic profile, sarcopenia, volume status, proteinuria, blood pressure control, and many others. Perhaps it is time to heed the wise from the past and stop equating the decline in a single laboratory parameter, even one as seemingly sophisticated as eGFR, with chronic progressive organ disease without carefully assessing the context in a given individual case.

## Figures and Tables

**Figure 1 toxins-14-00127-f001:**
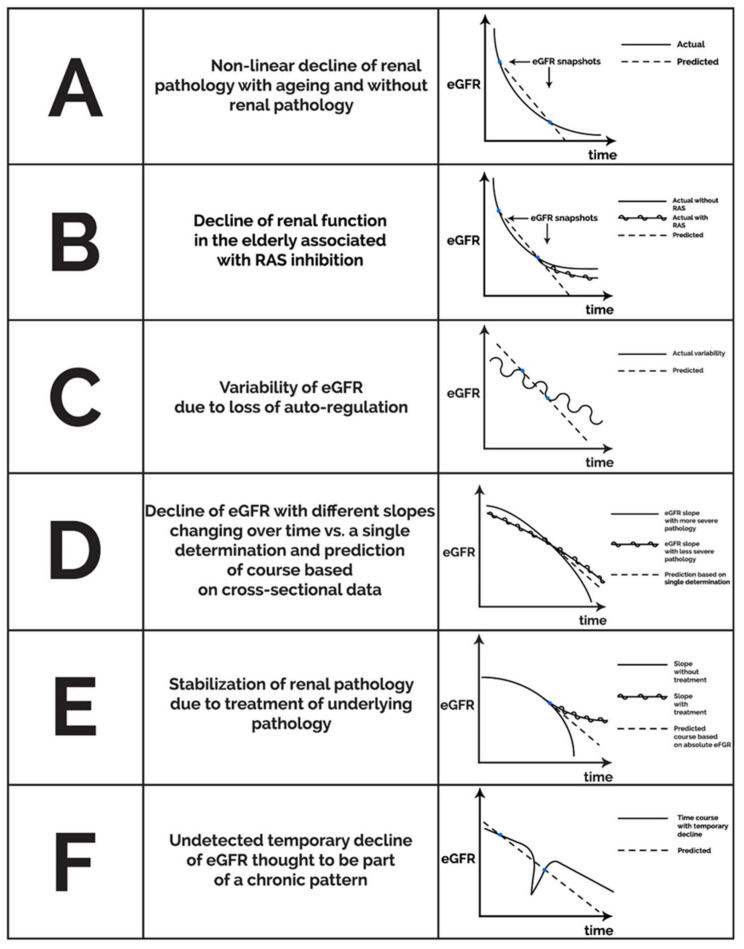
Potential theoretical patterns of non-linear eGFR decline unmasking a snapshot effect for infrequent eGFR determinations with linear extrapolation and pitfalls of basing risk on low absolute eGFR alone.

**Table 1 toxins-14-00127-t001:** Pitfalls of equating low estimated GFR with progressive renal pathology.

**Theoretical:**
-eGFR reflects only one of the many functions of the human kidney
-eGFR correlates only loosely with important predictors such as proteinuria, fluid status, blood pressure, metabolic acidosis, anemia, metabolic bone disease, iron deficiency, inflammation, tubular function
**Clinical:**
-eGFR has intrinsic day-to-day variability depending on dietary intake, cardiac output, fluid status, blood pressure, and medication use including RAS inhibitors
-eGFR has a unique non-linear pattern of decline with age and without renal pathology
-eGFR variability and slope may themselves be predictors of outcome
**Methodical:**
-under-represented populations when validating eGFR as a clinical marker
-variable correlation with clinical outcomes in certain glomerulopathies and diabetic kidney disease

## Data Availability

Not applicable.

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
