# Peer review of "Estimated Glomerular Filtration Rate in Chronic Kidney Disease: A Critical Review of Estimate-Based Predictions of Individual Outcomes in Kidney Disease"

_toxins, 2022, doi:10.3390/toxins14020127_

Round 1
Reviewer 1 Report
Well written review
Minor grammatical errors requiring correction
Author Response
Thank you for your kind assessment of our paper as a “well-written review”.
We addressed your suggestion to correct minor grammatical errors with an additional review by a native English speaker.

Reviewer 2 Report
Comment To Authors
The manuscript is interesting considering that in nephrology the use of eGFR alone and above all of the influence of age and other factors on it is still debated, in particular in the elderly.
-Casey M. Rebholz, Morgan E. Grams, Kunihiro Matsushita, Elizabeth Selvin, Josef Coresh. Change in Novel Filtration Markers and Risk of ESRD. Am J Kidney Dis. 2015 Jul; 66(1): 47–54.
-Andrew S. Levey, Lesley A. Inker, Josef Coresh. GFR Estimation: From Physiology to Public Health. Am J Kidney Dis. Am J Kidney Dis. 2014 May; 63(5): 820–834.
-Leslie A. Obert, Susan A. Elmore, Daniela Ennulat, Kendall S. Frazier. A Review of Specific Biomarkers of Chronic Renal Injury and Their Potential Application in Nonclinical Safety Assessment Studies. Toxicol Pathol. 2021 Jul; 49(5): 996–1023.
-Ahmed Alaini, Deepak Malhotra, Helbert Rondon-Berrios, Christos P Argyropoulos, Zeid J Khitan, Dominic S C Raj, Mark Rohrscheib, Joseph I Shapiro, Antonios H Tzamaloukas. Establishing the presence or absence of chronic kidney disease: Uses and limitations of formulas estimating the glomerular filtration rate. World J Methodol. 2017 Sep 26; 7(3): 73–92.
The authors' conclusions are certainly not conclusive but they can help clinicians to consider the complexity of the individual patient more than the eGFR alone.
…”Perhaps it is time to heed the wise from the past and stop equating the decline of a single laboratory parameter, even one as seemingly sophisticated as eGFR, with chronic progressive organ disease without carefully assessing the context in a given individual case”…
In particular ….”it is logical to assume that incidental measurements of eGFR without an assessment of the eGFR slope, proteinuria, blood pressure, age and co-morbidities should not establish the diagnosis of organic disease in the elderly population”…….
-Silvia Lai, Maria Ida Amabile, Silvia Altieri, Daniela Mastroluca, Carlo Lai, Paola Aceto, Massimiliano Crudo, Anna Rita D’Angelo, Maurizio Muscaritoli, Alessio Molfino. Effect of Underlying Renal Disease on Nutritional and Metabolic Profile of Older Adults with Reduced Renal Function. Front Nutr. 2017; 4: 4. Published online 2017 Mar 17. doi: 10.3389/fnut.2017.00004
however, in clinical practice it is necessary to have a simple, inexpensive and widespread parameter that can be used. In this sense the authors should give more suggestions or anyway reflections.
The manuscript is well written
Comment To Authors
The manuscript is interesting considering that in nephrology the use of eGFR alone and above all of the influence of age and other factors on it is still debated, in particular in the elderly.
-Casey M. Rebholz, Morgan E. Grams, Kunihiro Matsushita, Elizabeth Selvin, Josef Coresh. Change in Novel Filtration Markers and Risk of ESRD. Am J Kidney Dis. 2015 Jul; 66(1): 47–54.
-Andrew S. Levey, Lesley A. Inker, Josef Coresh. GFR Estimation: From Physiology to Public Health. Am J Kidney Dis. Am J Kidney Dis. 2014 May; 63(5): 820–834.
-Leslie A. Obert, Susan A. Elmore, Daniela Ennulat, Kendall S. Frazier. A Review of Specific Biomarkers of Chronic Renal Injury and Their Potential Application in Nonclinical Safety Assessment Studies. Toxicol Pathol. 2021 Jul; 49(5): 996–1023.
-Ahmed Alaini, Deepak Malhotra, Helbert Rondon-Berrios, Christos P Argyropoulos, Zeid J Khitan, Dominic S C Raj, Mark Rohrscheib, Joseph I Shapiro, Antonios H Tzamaloukas. Establishing the presence or absence of chronic kidney disease: Uses and limitations of formulas estimating the glomerular filtration rate. World J Methodol. 2017 Sep 26; 7(3): 73–92.
The authors' conclusions are certainly not conclusive but they can help clinicians to consider the complexity of the individual patient more than the eGFR alone.
…”Perhaps it is time to heed the wise from the past and stop equating the decline of a single laboratory parameter, even one as seemingly sophisticated as eGFR, with chronic progressive organ disease without carefully assessing the context in a given individual case”…
In particular ….”it is logical to assume that incidental measurements of eGFR without an assessment of the eGFR slope, proteinuria, blood pressure, age and co-morbidities should not establish the diagnosis of organic disease in the elderly population”…….
-Silvia Lai, Maria Ida Amabile, Silvia Altieri, Daniela Mastroluca, Carlo Lai, Paola Aceto, Massimiliano Crudo, Anna Rita D’Angelo, Maurizio Muscaritoli, Alessio Molfino. Effect of Underlying Renal Disease on Nutritional and Metabolic Profile of Older Adults with Reduced Renal Function. Front Nutr. 2017; 4: 4. Published online 2017 Mar 17. doi: 10.3389/fnut.2017.00004
however, in clinical practice it is necessary to have a simple, inexpensive and widespread parameter that can be used. In this sense the authors should give more suggestions or anyway reflections.
The manuscript is well written, some minor linguistic improvement may be needed
Author Response
Thank you for your valuable suggestions and your assessment of our paper as a manuscript “interesting” and “well-written”.
We incorporated all of your references extending the section entitled “The concept of the estimation of Glomerular Filtration Rate by estimating the clearance of a marker: the estimation of an estimation”.
Please see the highlighted text at the end of the chapter.
We also added “more reflections” on the potential clinical value of eGFR as part of the screening and diagnosis of CKD.
Please see the highlighted text in the section “Conclusions”.
Your suggestion for minor linguistic corrections is addressed with an additional review by a native English speaker.

Reviewer 3 Report
Presented manuscript is an excellent compilation of data related with GFR calculation and interpretation of its results, necessary especially for non-nephrologists. Mistakes related with GFR analysis lead often to unnecessary diagnosis of CKD, inappropriate therapeutic decisions or on the other hand, ignoring signs which may be related with patients worse outcomes. Unfortunately, lack of critical analysis of patient's condition, what has tremendous effect of GFR results, is still quite common clinical proble, what was emphasized in the manuscript. Another problem is proteinuria and its forgotten influence on patients mortality, improperly underestimated in comparison with GFR value. Chapter about GFR changes related with aging is another crucial part of the paper. Main points highlighted in the Table and variability of GFR changes shown in the Figure make the manuscript more useful in the decision making process.
However:
1) please unify abbreviations use, i.e. CKD is mentioned in first lines, whereas chronic kidney disease is used many times later in the text; similar situation is with ARBs; RBF is not explain in the text;
2) sartans are rather called 'angiotensin II type 1 receptor blockers' (ARB), not AIIRB (line 50), similarly, flozins are SGLT-2 inhibitors, not SGLT-1 (line 170) or SGT2i (line 372);
3) please unify references styles, some contain full names of Authors, some have 'et al.' after 2nd Author, number/dates are in wrong places, etc.
Author Response
Thank you for assessment of our paper as an “excellent compilation of data related with GFR calculation and interpretation of its results, necessary especially for non-nephrologists”.
We addressed your suggestions by:
- Correcting inconsistencies with abbreviations such as CKD, ARB, RBF and
- ARB and SGLT-2 inhibitors
- References have been completely reworked using EndNote
